# Simultaneous Measurement of Ear Canal Movement, Electromyography of the Masseter Muscle and Occlusal Force for Earphone-Type Occlusal Force Estimation Device Development

**DOI:** 10.3390/s19153441

**Published:** 2019-08-06

**Authors:** Mami Kurosawa, Kazuhiro Taniguchi, Hideya Momose, Masao Sakaguchi, Masayoshi Kamijo, Atsushi Nishikawa

**Affiliations:** 1Interdisciplinary Graduate School of Science and Technology, Shinshu University, 3-15-1 Tokida, Ueda, Nagano 386-8567, Japan; 2Graduate School of Engineering Science, Osaka University, 1–3 Machikaneyama, Toyonaka, Osaka 560-8531, Japan; 3Graduate School of Information Sciences, Hiroshima City University, 3-4-1 Ozukahigashi, Asaminami-ku, Hiroshima 731-3194, Japan; 4SKINOS Nagano Co., Ltd., 2-16-24 Fumiiri, Ueda, Nagano 386-0017, Japan; 5SKINOS Tohmi Lab., 3100-5 Kano, Tomi, Nagano 389-0505, Japan; 6Faculty of Textile Science and Technology, Shinshu University, 3-15-1 Tokida, Ueda, Nagano 386-8567, Japan

**Keywords:** occlusal force estimation, ear sensor, ear canal movement, optical measurement, masseter muscle myopotential, occlusal force, correlation, partial correlation

## Abstract

We intend to develop earphone-type wearable devices to measure occlusal force by measuring ear canal movement using an ear sensor that we developed. The proposed device can measure occlusal force during eating. In this work, we simultaneously measured the ear canal movement (ear sensor value), the surface electromyography (EMG) of the masseter muscle and the occlusal force six times from five subjects as a basic study toward occlusal force meter development. Using the results, we investigated the correlation coefficient between the ear sensor value and the occlusal force, and the partial correlation coefficient between ear sensor values. Additionally, we investigated the average of the partial correlation coefficient and the absolute value of the average for each subject. The absolute value results indicated strong correlation, with correlation coefficients exceeding 0.9514 for all subjects. The subjects showed a lowest partial correlation coefficient of 0.6161 and a highest value of 0.8286. This was also indicative of correlation. We then estimated the occlusal force via a single regression analysis for each subject. Evaluation of the proposed method via the cross-validation method indicated that the root-mean-square error when comparing actual values with estimates for the five subjects ranged from 0.0338 to 0.0969.

## 1. Introduction

A patient who undergoes a gastrectomy operation for gastric cancer is at increased risk of nutritional disorders because of their reduced gastric function [1]. It is thus important to improve their meal intake method to prevent such nutritional disorders. One way to improve the meal intake method is to ensure that the patient chews food well. Recently, a study that used the number of chews during eating for adiposity risk evaluation was conducted in Japan. We have previously performed research and development of a reliable earphone-type chewing-count measurement device (called the earable RCC) [2]. The “earable” part of the earable RCC name is a coined term that combines “wearable” with “ear”, while the “RCC” part is an abbreviation of “reliable chewing-count measurement device”. The earable RCC is a device that measures the number of chews performed by the user via an earphone-type sensor (ear sensor) [3] that we researched and developed to measure the movement of the ear canal. The device enables the total number of chews to be displayed on a tablet terminal in real time. In addition, it can also record the number of chews and the measured waveform on the tablet. The earable RCC is used for experimental analysis of the dietary behavior of adipose patients and to aid in provision of meal instructions for post-gastrectomy patients in a medical institution in Japan.

In addition, we have studied occlusal force measurement as another application that has been used successfully in measurement of mealtimes [4,5,6,7], respiratory rates [8], disturbances in breathing and posture during zazen [9], movements of the tongue [10], and movements of the eyes and intentional blinking [11] while using the same ear sensor from the earable RCC in our previous study. Our next action is thus to conduct research and development of an occlusal force estimation device based on this ear sensor. Previous occlusal force measurement systems required insertion of a pressure sensor into the patient’s mouth [12,13,14,15,16,17,18,19,20,21,22,23] or placement of electrodes on the patient’s jaw or cheek to measure the electromyography for occlusal force estimation [24,25,26,27,28,29].

In this research, we aim to develop an occlusal force measurement device using the ear sensor from the earable RCC to measure the number of chews. Such a device would mean that it would not be necessary to insert a sensor or a device into the patient’s mouth (i.e., it can perform measurements during eating) and does not require electrode pads, which can impede movement of the masticatory muscle and the jaw joint.

In advance of this study, we conducted an experiment involving five subjects about the correlation between the occlusal force with light chewing on second molar (occlusal force from approximately 0 to 40 N) and the movement of the ear canal measured using the ear sensor [4,30]. Using the results, the strong correlation between the occlusal force and the movement of the ear canal was then investigated.

However, the maximum occlusal force measured on the second molar was approximately 500 N to 600 N for healthy male subjects and approximately 400 N for a male subject with dentures [31]. Therefore, we performed a similar experiment with heavy chewing to confirm the correlation coefficient between the occlusal force and the ear sensor output over an appropriate occlusal force range.

This article presents the results of simultaneous measurement of the ear canal movement, the electromyography of the masseter muscle and the occlusal force (exceeding a maximum of 400 N), along with a discussion of each correlation based on investigation of the Pearson product-moment correlation coefficient and the partial correlation coefficient, due to push forward the study.

Section 2 describes the experimental system, the details of five subjects, the experimental method, the estimation method of the occlusal force, and the evaluation method for the estimation method. Section 3, Section 4 and Section 5 present the results, discussion, and conclusions, respectively.

## 2. Materials and Methods

### 2.1. Experimental System

In the experimental system shown in Figure 1, analog signals ranging from 0 V to 3.3 V measured using the earphone-type sensor that we developed to measure ear canal movement (ear sensor), the GM-10 occlusal force meter (Nagano Keiki Co., Nagano, Japan) and the BR-1000 electromyograph (Nishizawa Electric Meters Manufacturing Co., Ltd., Nagano, Japan) are converted into digital signals using an analog-to-digital (AD) converter at a sampling frequency of 100 Hz with 12 bit resolution. These digital signals are then recorded with timestamps using a storage device.

Figure 2 illustrates the principle of ear canal movement measurement using the ear sensor. Occlusion is performed by the temporalis and masticatory muscles, including the masseter muscle and the temporomandibular joint. Occlusion causes a change in the shape of the ear canal near the masticatory muscles and the temporomandibular joint. The ear sensor measures the change in the ear canal shape during occlusion optically and noninvasively.

A small QRE1113 photosensor (Fairchild Semiconductor International Inc., Sunnyvale, CA, USA) is attached to the ear sensor. The photosensor houses a light-emitting diode (LED) with an emission wavelength of 940 nm and a phototransistor, as illustrated in Figure 1. The ear sensor irradiates the skin of the ear canal with infrared light, and the reflected light is then received by the phototransistor to measure the change in the ear canal shape. In the ear sensor, the output increases as the amount of light reflected from the ear canal increases. Similarly, the output decreases as the amount of reflected light diminishes. The output offset voltage of the ear sensor can be adjusted using the variable resistor VR_1_. The LED is provided along with a pulse wave generator to control the light emission. This pulse wave generator is synchronized with the AD convertor that is connected to the ear sensor. Because of the mechanisms involved, light is only emitted during AD conversion, and it is thus possible to enhance the LED emission (i.e., to increase the LED’s forward current) when compared with the always-emitting case. As a result of the large quantity of light produced, the effects of any ambient infrared light transmitted via the skin near the outer ear can be suppressed and an improvement in the signal-to-noise (SN) ratio is expected. A processed medium-size commercial earplug, the EP3-BK-MPR (SureFire LLC., Fountain Valley, CA, USA), is used as a housing for the ear sensor.

The GM-10 occlusal force meter is a small and lightweight occlusal force meter, with width, height, depth (i.e., total length) and mass of 29 mm, 18 mm, 189 mm, and 70 g, respectively (see Figure 3). This occlusal force meter is constructed continuously from an oral cavity insert part and a gripping part; 88 mm of the total length is the oral cavity insert part (on the left side in Figure 3) and the remaining 101 mm forms the gripping part (on the right side in Figure 3). During measurements, the disposable resin-made cover is placed on the intraoral insertion part in advance. The sensor measures the occlusal force that acts during chewing of the tip of the intraoral insertion part when the gripping part is held using a single hand. The sensing area is located on the tip of the intraoral insertion part and propylene glycol liquid is sealed within its interior. The interior liquid pressure changes when the sensing area is chewed. The occlusal force meter then calculates the occlusal force from the liquid pressure, which is measured using a pressure sensor. The liquid pressure measurement and the occlusal force calculation are conducted by a signal processing circuit that is built into the gripping part. The calculated occlusal force is then displayed on a liquid crystal display device (on the back of the device as shown in Figure 3) that is built into the gripping part. The signal processing circuit consists of a microprocessor, a memory circuit, a timer circuit and a counter. The occlusal force meter used in this study was specially remodeled by its designers (coauthors Sakaguchi and Momose) for the research. Specifically, the remodeled occlusal force meter outputs the liquid pressure as the analog voltage *ν*_out_ V, which ranges from 0 V to 3 V. In the experimental system shown in Figure 1, *ν*_out_ is input by the AD convertor. The AD converted value can be converted into the occlusal force *F* N using Equation (1), which was calculated based on an experiment:*F =* 1985.6*ν*_out_ − 75.066(1)

The BR-1000 electromyograph device measures the surface electromyography (EMG) of the masseter muscle via the bipolar derivation method. The BR-1000 electromyograph consists of two detection electrodes, a ground electrode to determine the standard for the electric potential of the electromyograph, and a myogenic potential detection circuit. As an electrode, the Bioload 45352V (GE Healthcare, Chicago, IL, USA) was used. The EMG signal is bipolar and is led by the detection electrodes for input into the myogenic potential detection circuit. The myogenic potential detection circuit consists of a differential amplifier, a band-pass filter (BPF) with a passband of 100 Hz–1 kHz, an absolute value circuit, a low-pass filter (LPF) with a cut-off frequency of 100 Hz and a full-wave rectifier connected in cascade. This myogenic potential detection circuit outputs the envelope curve of a full-wave rectified signal in which the measured myogenic potential has been filtered and amplified. The electromyograph used in this study was specially remodeled by its designers (coauthors Sakaguchi and Momose) for this research. Specifically, the remodeled electromyograph outputs the envelope curve as the analog voltage *ν*_out_ V with a range from 0 V to 3 V. In the experimental system shown in Figure 1, the analog value is input as the EMG value by the AD convertor.

The AD convertor and the pulse wave generator shown in the experimental system in Figure 1 were implemented using an mbed LPC1768 microprocessor (switch science Inc., Tokyo, Japan) and proprietary software (written in C++). The AD convertor inputs from 0 V to 3.3 V are converted into values ranging from 0.0000 to 1.0000, respectively.

A personal computer (PC; CF-SV78SJQP, Panasonic Corp., Kadoma, Osaka, Japan) equipped with Windows 10 Pro Ver. 1803 (Microsoft Corp., Redmond, WA, USA) was used as the storage device. The microprocessor and the PC communicated using a CRS-232 specification interface. For this communication, the communication software CoolTermWin Ver. 1.4.7 (freeware) was used. The acquired data in the PC were processed using spreadsheet software (Microsoft Excel for Office 365, MOS 32-bit, Microsoft Corp., Redmond, WA, USA).

### 2.2. Subjects

The subjects were five young and healthy people (males and females, aged 21 to 26, with average age of 23.2 years), who were denoted by A, B, C, D and E. These subjects were able to chew the sensing area of the occlusal force meter thoroughly with their second molar teeth without any trouble and suffered no symptoms of pain or fatigue in the teeth and/or the jaw. People who were undergoing orthodontic treatment or being treated for a tooth or temporomandibular joint problem were excluded from the tests. In addition, subjects for whom the size of the ear sensor was a close fit to the size of their ear were selected. These people were free from symptoms of pain or fatigue in their ears. People undergoing treatment for ear problems were excluded. In addition, any person who felt discomfort such as itching or for whom the seal-type electrode pad for the EMG measurements caused inflammation was excluded. The study was conducted in accordance with the Declaration of Helsinki, and received the approval of the “Ethics Review Procedures concerning research with human subjects at Shinshu University (project identification code: 227)”; after the contents of the experiments were explained, each of the subjects signed a written consent.

### 2.3. Simultaneous Measurement of Ear Canal Movement, Masseter Muscle Electric Potential and Occlusal Force

To examine the correlation between the ear canal movement (from the ear sensor), the masseter muscle’s electric potential (from the EMG) and the occlusal force, the measured values from the ear sensor, the EMG, and the occlusal force meter were recorded together with timestamps when the subjects chewed the sensing area of the occlusal force meter thoroughly with their second molar teeth. Each of five subjects described in Section 2.2 performed the following experiment for six runs. In the experiment, a subject is seated in a chair with an ear sensor placed on their right ear to measure the movement of their ear canal; two surface electrodes for detection were placed on their right cheek (where the masseter muscle is located) to measure the EMG and a ground electrode was placed on their right clavicle. After putting on the ear sensor and the electrodes, the subject chews the sensing area of the occlusal force meter thoroughly with their second molar teeth. In the experiment, the subject gradually increases the pressure on the occlusal force sensor for approximately 2 s from the condition under which the sensing part of the occlusal force meter is chewed lightly to the maximum pressure in the 2 s to fix it in their mouth. Then, the pressure is relaxed to return to the initial conditions.

The subject kept the head still during the experiment to prevent motion artifacts originating from the motion of the head during the measurements of the ear canal.

For hygiene reasons, the ear sensor was cleaned using a clean brush and disinfected using ethanol before and after each experiment in all the experiments. The subjects cleaned their ear canal using a cotton swab before and after each experiment. The disposable cover of the occlusal force meter was replaced after every use and the body of the occlusal force meter was disinfected using ethanol after every use. In addition, the subjects gargled before and after each experiment. The disposable surface electrode of the EMG device was replaced for every subject. Before the electrodes were attached, the skin of the subject was wiped with ethanol as a skin preprocessing step.

### 2.4. Method of Occlusal Force Estimation from Ear Canal Movement Measured by Ear Sensor and Associated Evaluation Method

Among the six runs for each subject in Section 2.3, the outputs from both sensors in the five runs were used as a training set, and the remaining run was then used as a test set. The estimation accuracy was evaluated by K-fold cross-validation. For this study, the value of *K* is then six as six measurements were obtained. In this study, the precision of the *k* th run was calculated and it was used as a test set, while the remaining five runs were used as training sets. This was performed for *k* = 1, 2, …, *K*.

In our previous research [4,30], a strong correlation was found with regard to the Pearson product-moment correlation coefficient between the ear sensor outputs and the occlusal force for weak occlusal forces of approximately 40 N; we then successfully obtained a regression line using the least squares method. Therefore, we also used single regression analysis to perform the estimations in this study. Specifically, the occlusal force estimated by single regression analysis was calculated by following procedures I to IV (corresponding to Equations (2)–(4)). The estimation accuracy was examined using the estimation evaluation index given in Equation (5). Additionally, an experiment with a strong occlusal force of more than 400 N was performed in this study. We then discussed how enhancement of the occlusal force affects a specific correlation in Section 4 based on the experimental results.
Training set **T***_k_*= {**T***_k_*_1_, **T***_k_*_2_, **T***_k_*_3_, **T***_k_*_4_, **T***_k_*_5_} is composed of the measured results from the five runs, with the exception of the *k* th measured result (test set). This was performed for *k* = 1, 2, …, 6. The *i* th element of the training set was composed of the occlusal force meter output *f_ij_* (AD converted value) and the ear sensor output *e_ij_* (AD converted value) measured within 2 s at a sampling frequency of 100 Hz, and was used as **T*_ki_*** = { *e_ij_*, *f_ij_*|*j* = 1, 2, 3, …, 200}. *j* was the sequential number added to each data sample.Second item SLOPE is a function used to obtain the single regression coefficient *a_ki_* to estimate occlusal force from the ear sensor value, as shown in Equation (2). *a_ki_* was determined for every element of the training set **T***_ki_*:(2)$$a_{ki}=\mathrm{SLOPE}\left(T_{ki}\right)=\frac{200\sum_{j=1}^{200}e_{ij}f_{ij}-\left(\sum_{j=1}^{200}e_{ij}\right)\left(\sum_{j=1}^{200}f_{ij}\right)}{200\sum_{j=1}^{200}e_{ij}^{2}-{\left(\sum_{j=1}^{200}e_{ij}\right)}^{2}}\;k=\;1,\;2,\;3,\;4,\;5,\;6\;i=\;1,\;2,\;3,\;4,\;5$$The average $\bar{a_{k}}$ of the single regression coefficient *a_ki_* obtained for every element of the training set **T***_ki_* was determined using Equation (3):(3)$$\bar{a_{k}}=\frac{1}{5}\overset{5}{\underset{i=1}{\sum }}a_{ki}\;k=\;1,\;2,\;3,\;4,\;5,\;6$$Let *f_k0_* be the initial value of the occlusal force in the *k* th test, and *e_k_*_0_ be the initial value from the ear sensor in the *k* th test. The estimated value $\tilde{f}\;$*_kj_* was determined by substitution of the measured value from the ear sensor *e_k_*_j_ into Equation (4):(4)$${\tilde{f}}_{kj}={\bar{a}}_{k}\left(e_{kj}-e_{k0}\right)+f_{k0}\;k=\;1,\;2,\;3,\;4,\;5,\;6\;j=\;1,\;2,\;3,\;\ldots ,\;200$$

The precision evaluation index $NRMSE_{k}$ given in Equation (5) was used to evaluate the accuracy of the estimated value ${\tilde{f}}_{kj}$:(5)$$NRMSE_{k}=\frac{RMSE_{k}}{{\tilde{f}}_{k\mathrm{MAX}}-{\tilde{f}}_{k\mathrm{MIN}}}\;k=\;1,\;2,\;3,\;4,\;5,\;6$$

The root-mean-square error *RMSE**_k_* was determined using Equation (6), where ${\tilde{f}}_{k\mathrm{MAX}}$ and ${\tilde{f}}_{k\mathrm{MIN}}$ respectively indicate the maximum and minimum estimated values in ${\tilde{f}}_{k1}$, $\;{\tilde{f}}_{k2}$, ${\tilde{f}}_{k3}$,…, $\;{\tilde{f}}_{k200}$:(6)$$RMSE_{k}=\sqrt{\frac{1}{200}\overset{200}{\underset{j=1}{\sum }}{\left(f_{kj}-{\tilde{f}}_{kj}\right)}^{2}}\;k=\;1,\;2,\;3,\;4,\;5,\;6$$

## 3. Results

### 3.1. Simultaneous Measurement Results for Ear Canal Movement, Masseter Muscle and Occlusal Force for Earphone-Type Occlusal Force Sensor

The average and square root values of the unbiased variance from the Pearson product-moment correlation coefficient between the ear sensor value and the occlusal force, the ear sensor value and the EMG value and the EMG value and the occlusal force obtained in the six runs for subjects A through E using the experimental method described in Section 2.3 are given in Table 1. The square root of the unbiased variance was used to show the dispersion of the average in the correlation coefficients. When the variation of the average in a correlation coefficient decreases, the value of the square root of the unbiased variance also decreases in tandem. Based on the results presented in Table 1, the average and square root values of the unbiased variance of the partial correlation coefficient in the six runs when eliminating the effect of the EMG value from the correlation coefficient between the ear sensor value and the occlusal force are given in Table 2. The average and the square root of the unbiased variance of the partial correlation coefficient in the six runs when eliminating the effect of the ear sensor value from the correlation coefficient between the EMG value and the occlusal force are also presented in Table 2. When estimating the occlusal force with the ear sensor value given in Section 2.4, the absolute value of the average of the correlation coefficient and the partial correlation coefficient should be close to 1, while the square root of the unbiased variance should be close to 0.

As shown in Table 1, the correlation coefficient between the ear sensor value and the occlusal force and the correlation coefficient between the ear sensor value and the EMG value were negative for subjects A, B and D, but were positive for subjects C and E. The correlation coefficient between the EMG value and the occlusal force was positive for all subjects. The correlation coefficient can be a positive correlation or a negative correlation, depending on the specific subject. In addition, the subjects who showed positive correlations only showed positive correlations and the subjects who showed negative correlations only showed negative correlations in all six runs. The absolute average of the correlation coefficient between the ear sensor and the occlusal force for all subjects was more than 0.9514 and the square root of the unbiased variance was 0.0262 or less. The absolute average of the correlation coefficient between the ear sensor value and the EMG value for all subjects was more than 0.9082, while the square root of the unbiased variance was 0.0362 or less. In addition, the absolute average of the correlation coefficient between the EMG value and the occlusal force for all subjects was more than 0.9549, while the square root of the unbiased variance was 0.0378 or less. Subject A showed the highest absolute average for the correlation coefficient between the ear sensor value and the occlusal force; their average was 0.9941, the square root of the unbiased variance was 0.0039, their minimum absolute correlation coefficient in the six runs was 0.9875 and the corresponding maximum was 0.9974. Subject B showed the lowest absolute average for the correlation coefficient between the ear sensor value and the occlusal force; their average was 0.9514, the square root of the unbiased variance was 0.0262, their minimum absolute correlation coefficient in the six runs was 0.9170 and the maximum was 0.9861. Subject A showed the highest absolute average for the correlation coefficient between the ear sensor value and the EMG value; their average was 0.9802, the square root of the unbiased variance was 0.0052, their minimum absolute correlation coefficient in the six runs was 0.9721 and the maximum was 0.9844. Subject D showed the lowest absolute average for the correlation coefficient between the ear sensor value and the occlusal force; their average was 0.9082, the square root of the unbiased variance was 0.0362, their minimum absolute correlation coefficient in the six runs was 0.8632 and the maximum was 0.9555. Subject B showed the highest absolute average for the correlation coefficient between the EMG value and the occlusal force; their average was 0.9832, the square root of the unbiased variance was 0.0060, their minimum absolute correlation coefficient in the six runs was 0.9736 and the maximum was 0.9903. Subject D showed the lowest absolute average for the correlation coefficient between the ear sensor value and the occlusal force; their average was 0.9549, the square root of the unbiased variance was 0.0197, their minimum absolute correlation coefficient in the six runs was 0.9290 and the maximum was 0.9785.

As shown in Table 2, the partial correlation coefficient between the ear sensor value and the occlusal force and the partial correlation coefficient between the ear sensor value and the EMG value were negative for subjects A, B and D, but were positive for subjects C and E. The partial correlation coefficient between the EMG value and the occlusal force was positive for all subjects. The relationships for dependence of negative or positive partial correlation coefficient values on the specific subjects were the same as those for the correlation coefficient. Subject A showed the highest absolute average for the partial correlation coefficient between the ear sensor value and the occlusal force; their average was 0.8286, the square root of the unbiased variance was 0.1532, their minimum absolute partial correlation coefficient in the six runs was 0.5262 and the maximum was 0.9393. Subject B showed the lowest absolute average for the partial correlation coefficient between the ear sensor value and the occlusal force; their average was 0.6161, the square root of the unbiased variance was 0.1860, their minimum absolute partial correlation coefficient in the six runs was 0.2904 and the maximum was 0. 8681. Subject B showed the highest absolute average for the partial correlation coefficient between the EMG value and the occlusal force; their average was 0.8551, the square root of the unbiased variance was 0.1029, their minimum absolute partial correlation coefficient in the six runs was 0.7188 and the maximum was 0.9522. Subject A showed the lowest absolute average for the partial correlation coefficient between the EMG value and the occlusal force; their average was 0.3501, the square root of the unbiased variance was 0.2857, their minimum absolute partial correlation coefficient in the six runs was 0.1215 and the maximum was 0.6325. The partial correlation coefficients between the ear sensor value and the occlusal force were higher than the partial correlation coefficients between the EMG value and the occlusal force for all subjects.

Figure 4 presents the measured results for subject A. The graph shows the measured results in which the correlation coefficient between the ear sensor value and the occlusal force was at its highest in the six runs of subject A. This measured value also showed the highest correlation between the ear sensor value and the EMG value. Here, the horizontal axis represents time, the left vertical axis represents the ear sensor value (i.e., the AD converted value), and the right vertical axis represents the EMG value (the AD converted value) and the occlusal force (the AD converted value). As indicated by Table 1 and Table 2, subject A had characteristics of the highest absolute average for the correlation coefficient between the ear sensor value and the occlusal force among all subjects; the highest absolute average for the correlation coefficient between the ear sensor value and the EMG value among all subjects; the second highest average for the correlation coefficient after subject B; the highest absolute average for the partial correlation coefficient between the ear sensor value and the occlusal force in all subjects; and the lowest average for the correlation coefficient between the EMG and the occlusal force among all subjects. Figure 4 also provides an overview of these characteristics.

### 3.2. Results for Estimation of Occlusal Force from Ear Canal Movement (Ear Sensor Value)

The average $\bar{a}$ from $\bar{a_{k}}$ (*k* = 1, 2, …, 6) as determined using Equation (3) and the square root of the unbiased variance for each subject are given in Table 3.

In Table 3, subject A showed the lowest square root of the unbiased variance in all subjects, with a value of 1.7487. The measurement results for the ear sensor and the occlusal force in the first through sixth runs for subject A are given in Figure 5. Here, the horizontal axis represents the ear sensor-measured value, while the vertical axis represents the measured occlusal force value. The trends in the six runs all showed the same inclinations with small differences in the offset values of the ear sensor values. Figure 5 showed intuitively that the square root of the unbiased variance for subject A can be small, as indicated by the results in Table 3. Subject B showed the highest square root of the unbiased variance among all subjects, with a value of 11.0181. The dispersion of $\bar{a_{k}}$ for subject B was greater than that for the other subjects.

Table 4 shows the results of cross-validation with regard to the estimated occlusal force from the ear sensor value Equation (4) using the method described in Section 2.4. Specifically, the *RMSE**_k_* values for subjects A through E determined using Equation (6), the maximum estimated value ${\tilde{f}}_{k\mathrm{MAX}}$, the minimum estimated value ${\tilde{f}}_{k\mathrm{MIN}}$ the difference between ${\tilde{f}}_{k\mathrm{MAX}}$ and ${\tilde{f}}_{k\mathrm{MIN}}$ (estimated width) and the average of the cross-validation results over six runs for the calculated results for the precision evaluation index $NRMSE_{k}$, as defined in Equation (5), are shown. Higher values for the estimated width in Table 4 represent wider estimation ranges. A lower *RMSE_k_* represents higher estimation accuracy for the same estimated width. Because the estimated width differed depending on the specific subject, a precision evaluation index $NRMSE_{k}$ in which the effect of the estimated width was eliminated was used for the evaluation. A lower $NRMSE_{k}$ value corresponds to higher evaluation accuracy. Subject A showed the highest accuracy (i.e., the lowest value of $\bar{NRMSE}$), with $\bar{RMSE}$ of 0.0338, an estimated width ($\bar{{\tilde{f}}_{\mathrm{MAX}}-{\tilde{f}}_{\mathrm{MIN}}}$) of 0.5390, and an $\bar{NRMSE}$ value of 0.0659. The estimated values for subject A were translated into occlusal forces using Equation (1) and gave values for $\bar{{\tilde{f}}_{\mathrm{MAX}}}$ of 1117 N and $\bar{{\tilde{f}}_{\mathrm{MIN}}}$ of 47 N. Subject B showed the highest value for $\bar{NRMSE}$, with $\bar{RMSE}$ of 0.0969, an estimated width of 0.3963, and an $\bar{NRMSE}$ value of 0.2869. The estimated values for subject B were translated into occlusal forces using Equation (1) and gave values for $\bar{{\tilde{f}}_{\mathrm{MAX}}}$ of 801 N and $\bar{{\tilde{f}}_{\mathrm{MIN}}}$ of 14 N.

Subject A showed the largest estimated width and subject C showed the smallest estimated width, where the latter is 0.2040.

## 4. Discussion

In related studies concerning the estimation of the occlusal force, one active area of investigation involves using EMG. In research on the correlation between the EMG and the occlusal force relevant to this study, a value for the correlation coefficient of 0.75 was given in [26] and 0.626 in [28]. The measurement device that we used for EMG gave 0.9549 or more for the correlation coefficient between the EMG and the occlusal force. These values of the correlation coefficient from these devices are not comparable because they have differences in structure and data processing techniques. Nevertheless, a correlation between the EMG and the occlusal force has been performed with the three devices and therefore we believe comparing our EMG measurement results as well as the measurement results of the ear sensor offers a means to contrast related research indirectly.

In previous research [4,30], a strong correlation between the ear sensor outputs and the occlusal force for weak occlusal forces of approximately 40 N had been investigated. In this research, we performed experiments with strong occlusal forces exceeding 400 N. Therefore, as shown in Table 1, each absolute average of the correlation coefficient between the ear sensor value and the occlusal force, the correlation coefficient between the ear sensor value and the EMG value, and the correlation coefficient between the EMG value and the occlusal force showed high correlation for all subjects. Additionally, the dispersion of the correlation coefficient was low, as indicated by the fact that the square root of the unbiased variance was low. Although the number of subjects was small, these considerations are roughly possible because the five subjects, whose ages and sex were different, performed the measurement results in a consistent manner. However, in order to obtain statistically significant results, it is essential to increase the number of subjects, and in the future, we will increase the number of subjects of various types.

The reason for the existence of both positive and negative correlation coefficients, as indicated by Table 1, can be attributed to inter-individual differences in ear canal shape. The diameter, length and bending of the ear canal can all vary. Because the ear sensor emits light toward the ear canal and outputs a voltage converted from the intensity of the reflected light, inter-individual differences such as bending of the ear canal affect the intensity of the reflected light directly. The partial correlation coefficient exceeded 0.6161 for all subjects, indicating that there is a correlation, as shown in Table 2. In Table 1, the correlation coefficient between the ear sensor and the occlusal force was as high as the correlation coefficient between the EMG and the occlusal force. Moreover, Table 2 suggested the correlation between the ear sensor and the occlusal force was established independently of EMG. Using the correlation coefficient and the partial correlation coefficient, the correlation between the ear sensor value and the occlusal force can be investigated; this represents an adequate method to acquire the estimated occlusal force from the value measured by the ear sensor via single regression analysis. Additionally, the low *RMSE* and *NRMSE* values from the experimental results also supported the appropriateness of the method. To improve the estimation accuracy, both collecting the measurement values of the ear canal movement from a subject of various characteristics by increasing the number of subjects and increasing the amount of learning data (i.e., for the training set) for the estimation process are valid approaches. However, the additive average value of $a_{ki}$ is used for the $\bar{a_{k}}$ value at present. To achieve improved estimation accuracy, we must consider other methods such as acquisition of a median and a mode from a histogram formed with increased quantities of learning data. Additionally, we will improve the estimation method and develop a data processing technique for the ear sensor with fewer errors and high robustness.

In this experiment, the value of the ear sensor changed proportionally with increasing occlusal force and had not attained a state of saturation for all subjects (Figure 5). However, we speculate that the ear sensor value attains a state of saturation for an occlusal force outside the range applied in the experiment. In such instances, ear canal movements may be hindered by an anatomical factor of the muscle and the internal skeleton.

This study was conducted with the supposition that the ear sensor does not misread a measurement. From the square root of the unbiased variance (Table 3) and the value of $\bar{NRMSE}$ in Equation (4), this supposition appears to hold in the range of the measurement results, because there was no measurement error troubling the estimation method indicated in Equation (4). However, there is concern that, with the movement of the head, a motion artifact may prompt mistakes in measurement. Because commercial earplugs were used to house the ear sensor. The earplugs do not fit perfectly for each subject. Therefore, the sensor may shake from movement of the subject’s head, and the shaking may affect measurement values. Mitigation of these concerns is an open problem for the future, as is perfect earplug fitting for each subject, which may help to eliminate motion artifact from measurement results.

Finally, we describe a future scenario for developing an occlusal force estimation system (referred to as “earable Ω”) with the estimation method in Section 2.4 (estimation method using the measured value from ear sensor). The “OMEGA” of the name “earable Ω” is an abbreviation of “Occlusal Measurement and its Estimation system, General Availability version”. The earable Ω comprising a measurement start button, a calibration button, and a small liquid crystal display device, is an ear phone shaped device including ear sensor. The earable Ω is put on the ear on the same side of the teeth used for measurement. The measured values from the ear sensor are recorded in memory with the digital signals converted by the AD converter at a sampling frequency of 100 Hz and resolution of 12 bit. Not only are measured results recorded but also $\bar{a_{k}}$, both of which are needed in advanced to estimate the occlusal force using the estimation method in Section 2.4. Only the maximum occlusal force on the loading small liquid crystal needs to be displayed by the earable Ω device. A display of its time-series variation can be had by sending the data to an external device such as a smart phone, from which the data can be further processes.

To obtain a training set to establish $\bar{a_{k}}$, gummy candy, which has a known hardness (adjusted hardness) is to be used instead of the occlusal force meter as in this study. Once the calibration button is activated, then the gummy candy is chewed with the teeth used for the measurement. The measured result of the first “single bite” is used for the training set. Repeated chewing of several gummy candies increases the data needed for $\bar{a_{k}}$ and improves the estimation accuracy.

When performing occlusal force estimations, the user places the earable Ω on the ear and brings the top back tooth and the bottom back tooth into contact without chewing (the occlusal force is then 0 N). The user next activates the start button of the earable Ω and chews. For the two-second duration from activation, measurements are recorded in memory. The occlusal force is then estimated using Equation (4) based on these measurements. The measured value from the ear sensor on activation is used as the initial value for $e_{k0}$ with $f_{k0}$ set to 0.

As a next step, we shall verify the idea by making a prototype of the earable Ω. The required specification necessary to estimate the occlusal force is still being studied along with other research organizations. In future, clinical experiments need to be undertaken to help determine suitable specifications regarding resolution and tolerance of the device.

## 5. Conclusions

We are engaged in research and development of a method for occlusal force estimation based on the movement of the ear canal and a device that uses the proposed method. The occlusal force can be measured during eating if it is possible to estimate the occlusal force from the movement of the ear canal. The method does not use electrode pads, which impede the movement of the masticatory muscle and the jaw joint. An earphone that we originally researched and developed was used to measure the ear canal movement. The ear sensor has the same shape as an internal-type earphone and contains an infrared LED and a phototransistor. The LED irradiates the inside of ear canal with light and the reflected light is received by the phototransistor to measure the change in the ear canal shape. In the ear sensor, the output increases in tandem with the amount of light reflected from the ear canal. Similarly, the output decreases in tandem with the amount of light reflected from the ear canal. The output offset voltage of the ear sensor can be adjusted using the variable resistor VR_1_. In this work, we simultaneously measured the movement of the ear canal, the surface electromyography (EMG) of the masseter muscle and the occlusal force six times each for five subjects as a basic study for the development of an occlusal force meter. We used these results to investigate the Pearson product-moment correlation coefficient between the ear sensor value and the occlusal force, and the partial correlation coefficient between the ear sensor values and the occlusal force when eliminating an effect of the EMG. Additionally, we investigated the average of the partial correlation coefficient over the six runs and the absolute value of the average for each subject. The results for the absolute value were indicative of strong correlation, with the correlation coefficients exceeding 0.9514 for all subjects. The lowest partial correlation coefficient shown by any subject was 0.6161, while the highest partial correlation coefficient shown by any subject was 0.8286. These values were also indicative of correlation. We then estimated the occlusal force via single regression analysis using the data from the six runs for each of the subjects. There is a correlation between the ear sensor value and the occlusal force that can be investigated that provides an adequate method to estimate the occlusal force from the ear sensor value via single regression analysis. The results of evaluation of the proposed method using the cross-validation method indicated that the root-mean-square error (*RMSE_k_*) obtained from comparison of the actual value with the estimates for the five subjects ranged from 0.0338 to 0.0969.

In future work, we intend to realize an occlusal force estimation device through improvement of the estimation accuracy by increasing the numbers of learning data, development of a data processing technique with fewer errors and high robustness that is suitable for use with the ear sensor, and improvement of the estimation method.

## Figures and Tables

**Figure 1 sensors-19-03441-f001:**
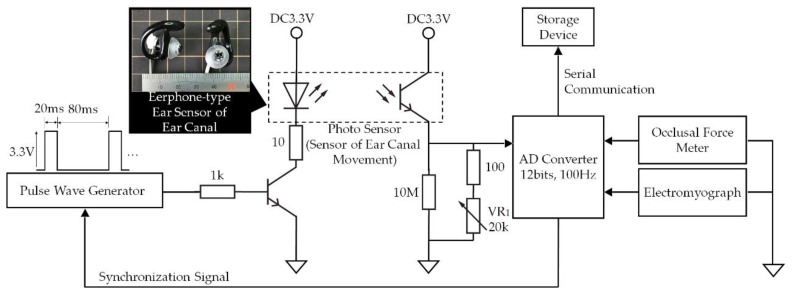
Experimental system. In this system, analog signals ranging from 0 V to 3.3 V measured using occlusal force meter, electromyography and the ear sensor to measure the movement of the ear canal are converted into digital signals by the analog-to-digital converter at a sampling frequency of 100 Hz with 12 bit resolution; the digital signals are then recorded together with timestamps in a storage device.

**Figure 2 sensors-19-03441-f002:**
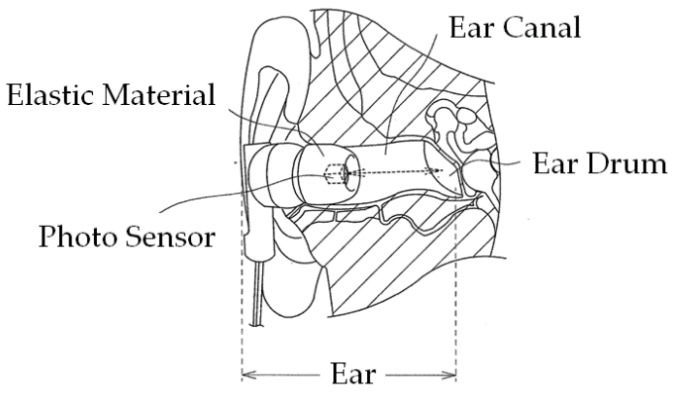
Principle of ear canal movement measurement using ear sensor. Occlusion is performed by the temporalis and masticatory muscles, including the masseter muscle and the temporomandibular joint. Occlusion causes a change in the ear canal shape near the masticatory muscles and the temporomandibular joint. The ear sensor measures this shape change in the ear canal during occlusion optically and noninvasively. A small photosensor is attached to the ear sensor. This photosensor houses a light-emitting diode (LED) with an emission wavelength of 940 nm and a phototransistor, as illustrated in Figure 1. The ear sensor irradiates the skin of the ear canal with infrared light, and the reflected light is then received by the phototransistor to measure the change in the ear canal shape.

**Figure 3 sensors-19-03441-f003:**
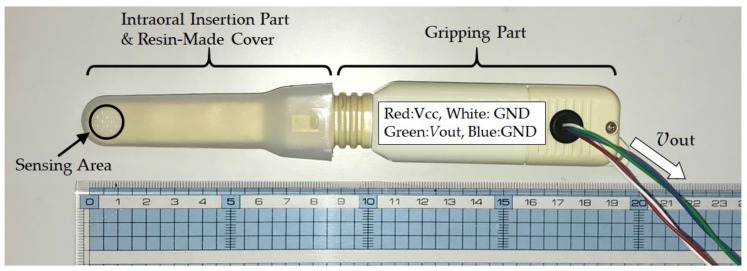
Appearance of the GM-10 occlusal force meter. This occlusal force meter is constructed continuously of an intraoral insertion part and a gripping part; 88 mm of the total length is the intraoral insertion part (on the left side in the figure) and the remaining 101 mm is the gripping part (on the right side in the figure). During measurements, the disposable resin-made cover is placed on the intraoral insertion part in advance. The subject then holds the gripping part using a single hand and the sensor measures the occlusal force when the subject chews the tip (i.e., the sensing area) of the intraoral insertion part.

**Figure 4 sensors-19-03441-f004:**
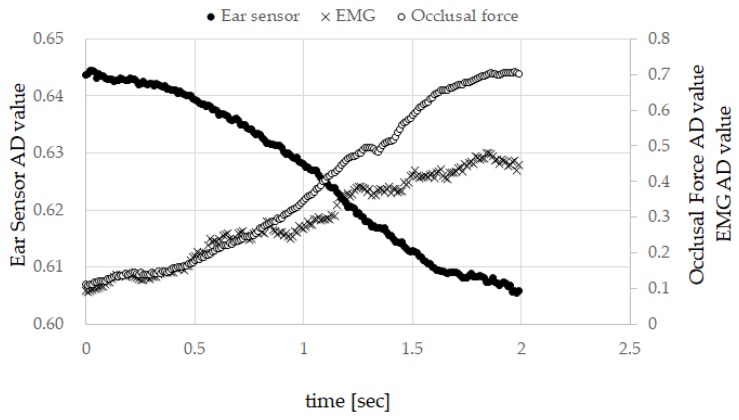
Measured results for subject A. The graph shows the measured results for which the correlation coefficient between the ear sensor value and the occlusal force was the highest among the six runs of subject A.

**Figure 5 sensors-19-03441-f005:**
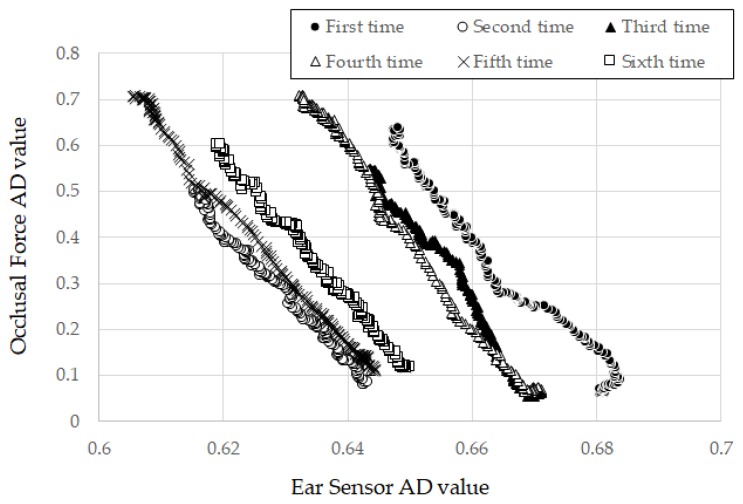
Measurement results for ear sensor and occlusal force over the first through sixth runs for subject A. Here, the horizontal axis represents the ear sensor-measured value, while the vertical axis represents the measured occlusal force value.

**Table 1 sensors-19-03441-t001:** Results for correlation coefficients. Average and square root values of unbiased variance from correlation coefficients between ear sensor value and occlusal force, ear sensor value and electromyography (EMG) value, and EMG value and occlusal force, as obtained over six runs for subjects A through E using the experimental method of Section 2.4 are given.

Subject	Ear Sensor—Occlusal Force		Ear Sensor—EMG		EMG—Occlusal Force	
	Average	Square Root of Unbiased Variance	Average	Square Root of Unbiased Variance	Average	Square Root of Unbiased Variance
A	−0.9941	0.0039	−0.9802	0.0052	0.9828	0.0059
B	−0.9514	0.0262	−0.9247	0.0319	0.9832	0.0060
C	0.9897	0.0035	0.9788	0.0119	0.9792	0.0134
D	−0.9597	0.0251	−0.9082	0.0362	0.9549	0.0197
E	0.9806	0.0072	0.9576	0.0255	0.9633	0.0378

**Table 2 sensors-19-03441-t002:** Results for partial correlation coefficients. Based on the results given in Table 1, the average and square root values of the unbiased variance of the partial correlation coefficient over six runs while eliminating the effect of the EMG value from the correlation coefficient between the ear sensor value and the occlusal force are given; the average and square root of the unbiased variance of the partial correlation coefficient over six runs while eliminating the effect of the ear sensor value from the correlation coefficient between the EMG value and the occlusal force are also given.

Subject	Ear Sensor—Occlusal Force		EMG—Occlusal Force	
	Average	Square Root of Unbiased Variance	Average	Square Root of Unbiased Variance
A	−0.8286	0.1532	0.3501	0.2857
B	−0.6161	0.1860	0.8551	0.1029
C	0.7282	0.1323	0.3668	0.2773
D	−0.7568	0.1691	0.6385	0.3023
E	0.6276	0.3290	0.4824	0.4846

**Table 3 sensors-19-03441-t003:** Average estimation results for the single regression coefficient and the square root of the unbiased variance. The average $\bar{a}$ from $\bar{a_{k}}$ (*k* = 1, 2,…, 6) determined using Equation (3) and the square root of the unbiased variance of $\bar{a_{k}}$ for each subject are given.

Subject	$\;\bar{a}\;$	Square Root of Unbiased Variance
A	−16.1064	1.7487
B	−22.8559	11.0181
C	12.0029	5.0236
D	−8.9634	3.7719
E	18.7544	5.2621

**Table 4 sensors-19-03441-t004:** Average precision error and maximum and minimum for each subject. Here, the precision errors *RMSE_k_* of subjects A through E determined using Equation (6) are shown, along with the maximum estimated value ${\tilde{f}}_{k\mathrm{MAX}}$, the minimum estimated value ${\tilde{f}}_{k\mathrm{MIN}}$, the difference between ${\tilde{f}}_{k\mathrm{MAX}}$ and ${\tilde{f}}_{k\mathrm{MIN}}$ (i.e., the estimated width), and the average of the cross-validation results over six runs for the calculated results for the precision evaluation index $NRMSE_{k}$ defined in Equation (5).

Subject	$\;\bar{RMSE}\;$	$\;\bar{{\tilde{f}}_{MAX}}\;$	$\;\bar{{\tilde{f}}_{MIN}}\;$	$\;\bar{{\tilde{f}}_{MAX}-{\tilde{f}}_{MIN}}\;$	$\;\bar{NRMSE}\;$
A	0.0338	0.6004	0.0614	0.5390	0.0659
B	0.0969	0.4411	0.0448	0.3963	0.2869
C	0.0489	0.2584	0.0544	0.2040	0.2670
D	0.0482	0.2699	0.0536	0.2163	0.2377
E	0.0915	0.4466	0.0176	0.4290	0.2078

## Data Availability

The measurement data used to support the findings of this study are restricted by the Shinshu University Ethics Committee for Studies Aimed at Humans in order to protect subject privacy. Data are available from authors for researchers who meet the criteria for access to confidential data.

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
