# Peer review of "Simultaneous Measurement of Ear Canal Movement, Electromyography of the Masseter Muscle and Occlusal Force for Earphone-Type Occlusal Force Estimation Device Development"

_sensors, 2019, doi:10.3390/s19153441_

Round 1
Reviewer 1 Report
The topic of the paper is interesting. The paper itself exhibits some interesting ideas.
The Abstract is concise and provides to the reader an adequate background of the study, defines its major purpose and describes how the study was performed.
The Introduction addresses the problem statement properly and provides a concise and elucidated background that places the reader in the study’s context. In addition, the authors clearly define what is the paper's main issue and briefly explain their approach, which is useful to contextualize and prepare the reader for the following sections.
The Materials and Methods section is well structured, well understood and properly outlines how the study was performed. The outcomes are clearly explained and supported with tables and figures that ease its understanding. The paper provides simple and well-understood discussion and conclusions.
However, the paper exhibits also a few problems. One of them, and probably the main one, is that of the number of subjects in the experiment. Only six subjects seem to be insufficient to get significant results. Subjects with more diverse features may be advised as it would conduct to a more robust study.
Another problems has to do with the premisses in which the study is supported. It is assumed that the sensors do not produce mistakes. Are there sensor models for the sensors used, i.e., are there information on the reliability of the values registered? What’s the probability of getting wrong measures? This is an important premisse to assess the results obtained, because the data set is built on them.
Yet another problem is that the authors fail to confront and compare their approach and results with other related research works, which is fundamental to prove the value and contributions of the research carried out for the scientific community.
Minor comments:
- The introduction should end with a paragraph describing the structure of the paper.
- Can’t understand the following sentence: “ In this study, K of K-fold cross-validation was six because measured result was split into with deeming one run of six run’s output runs being deemed one set. “
Overall, I think the paper is not mature enough for publication in this Journal.
I think the authors have implemented an interesting system, but, for the moment, they still need to express better the limitations of the paper, and improve the validation.
Author Response
We wish to express our appreciation to the reviewers for their insightful comments on our paper. The comments have helped us significantly improve the paper.
Point 1: Only six subjects seem to be insufficient to get significant results. Subjects with more diverse features may be advised as it would conduct to a more robust study.
Response 1: We have inserted two passages addressing the comment in the Discussion section (lines 416–418, 432–435).
Point 2: It is assumed that the sensors do not produce mistakes. Are there sensor models for the sensors used, i.e., are there information on the reliability of the values registered? What’s the probability of getting wrong measures? This is an important premisse to assess the results obtained, because the data set is built on them.
Response 2: Passages addressing the comment has been inserted in the Discussion section (lines 445-454).
Point 3: Yet another problem is that the authors fail to confront and compare their approach and results with other related research works, which is fundamental to prove the value and contributions of the research carried out for the scientific community.
Response 3: The comment is addressed with the insertion of a passage in the Discussion section (lines 400–408).
Point 4: The introduction should end with a paragraph describing the structure of the paper.
Response 4: An outline of the paper has been inserted in the Introduction section (lines 81–83).
Point 5: Can’t understand the following sentence: “In this study, K of K-fold cross-validation was six because measured result was split into with deeming one run of six run’s output runs being deemed one set.”
Response 5: We have rewritten the sentence that is indicated to clarify the misunderstanding (lines 222–223).
Point 6: Overall, I think the paper is not mature enough for publication in this Journal.
Response 6: We have revised the manuscript entirely in light of the comment of the two reviewers. The revised parts are in red and have been underlined.
Point 7: I think the authors have implemented an interesting system, but, for the moment, they still need to express better the limitations of the paper, and improve the validation.
Response 7: The system describe in this paper is still being researched and the final version has not been completed. System improvements and a greater number of subjects are necessary for system implementation, as has be indicated. Additionally, we have added an idea for system implementation in the Discussion section (lines 455–482) based on the above comment.

Reviewer 2 Report
The paper present results of a more extensive study from earlier work of the authors. I only have a few minor remarks.
- The optical recording method is comparable with PPG recording. I wonder if the authors could detect heartrate?
- How robust is this whole measurement setup to motion artefacts? I would think that any tiny head motion will impact the ear canal recording significantly? Some additional clarification, also in the recording protocol on how these head movements were controlled, would be needed,
- The correlation coefficient is very subject dependent. This is indicated by the fact that some have a positive and some have a negative correlation components. Since only 6 people were involved int he study, I think the group is not statistically significant. I wonder what the actual spread of correlation over a wider population would be? If it is positive for some and negative for others, could it not be that it will be actually 0 or close to 0 for others?
- One of the main differences with the earlier study, is the maximum force which is in this study much larger than the 40N from the previous study. I wonder if the correlation coefficient remains constant over the full range of force? Is the correlation sufficiently linear across the full force range? Or is some kind of saturation effect happening for very large forces exerted? I can imagine the ear canal cannot move too much? Some more discussion in this topic would be needed.
- Because of the large dependency on the subject, I wonder how this recording of ear canal movement could eventually be used as a predictive metric for the force? If for one subject the correlation is positive and for the other negative, how can this work in the final application scenario? It would need some kind of per subject calibration?
Author Response
We wish to express our appreciation to the reviewers for their insightful comments on our paper. The comments have helped us significantly improve the paper.
Point 1: The optical recording method is comparable with PPG recording. I wonder if the authors could detect heartrate?
Response 1: We have not tried heart rate detection. However, the comment has prompted us to try this out.
Point 2: How robust is this whole measurement setup to motion artefacts? I would think that any tiny head motion will impact the ear canal recording significantly? Some additional clarification, also in the recording protocol on how these head movements were controlled, would be needed.
Response 2: We have addressed the comment in the Experimental method section and the Discussion section (lines 209–210, 445–454).
Point 3: The correlation coefficient is very subject dependent. This is indicated by the fact that some have a positive and some have a negative correlation components. Since only 6 people were involved int he study, I think the group is not statistically significant. I wonder what the actual spread of correlation over a wider population would be? If it is positive for some and negative for others, could it not be that it will be actually 0 or close to 0 for others?
Response 3: We have addressed this problem in the Discussion section (lines 416–418). In the range of the experiment conducted in this study, there is no instance for which the correlation coefficient approaches 0. However, we can verify this point by improving the measurement value of the ear canal movement for each subject and with various characteristics by increasing the number of subjects (lines 432–435).
Point 4: One of the main differences with the earlier study, is the maximum force which is in this study much larger than the 40N from the previous study. I wonder if the correlation coefficient remains constant over the full range of force? Is the correlation sufficiently linear across the full force range? Or is some kind of saturation effect happening for very large forces exerted? I can imagine the ear canal cannot move too much? Some more discussion in this topic would be needed.
Response 4: In the range of the experiment in this study, the value of the ear sensor did not reach a state of saturation (see comment on lines 440–444). We also conducted a comparison with EMG as a difference from the previous research in this study (see comment on lines 400–408).
Point 5: Because of the large dependency on the subject, I wonder how this recording of ear canal movement could eventually be used as a predictive metric for the force? If for one subject the correlation is positive and for the other negative, how can this work in the final application scenario? It would need some kind of per subject calibration?
Response 5: A passage has been inserted in the Discussion section (lines 455–482) addressing the comment.

Round 2
Reviewer 1 Report
The authors have improved the paper and most of the issues were solved. However, I'm not yet convinced with the explanation given by the authors for the problem of having only five subjects in the study ("Although the number of subjects was small, these considerations are sufficiently possible because the five subjects, whose ages and sex were different, performed the measurement results in a consistent manner."). I still believe the only solution is increasing the number of subjects in order to get significant results.
Author Response
Point 1: I'm not yet convinced with the explanation given by the authors for the problem of having only five subjects in the study ("Although the number of subjects was small, these considerations are sufficiently possible because the five subjects, whose ages and sex were different, performed the measurement results in a consistent manner."). I still believe the only solution is increasing the number of subjects in order to get significant results.
Response 1: We have addressed the comment in the Discussion section (lines 417–420).
